**www.cambridge.org/ext**

# Selectivity of mass extinctions: Patterns, processes, and future directions

Jonathan L. Payne[1] , Jood A. Al Aswad[1] , Curtis Deutsch[2] ,
Pedro M. Monarrez[1] , Justin L. Penn[2] and Pulkit Singh[1]

[1]Department of Earth and Planetary Sciences, Stanford University, Stanford, CA, USA and [2]Department of Geosciences, Princeton University, Princeton, NJ, USA

biodiversity; climate; extinction; ecology; Physiology

**Corresponding author:**
Jonathan L. Payne;
Email: jlpayne@stanford.edu

## Abstract

A central question in the study of mass extinction is whether these events simply intensify background extinction processes and patterns versus change the driving mechanisms and associated patterns of selectivity. Over the past two decades, aided by the development of new fossil occurrence databases, selectivity patterns associated with mass extinction have become increasingly well quantified and their differences from background patterns established. In general, differences in geographic range matter less during mass extinction than during background intervals, while differences in respiratory and circulatory anatomy that may correlate with tolerance to rapid change in oxygen availability, temperature, and pH show greater evidence of selectivity during mass extinction. The recent expansion of physiological experiments on living representatives of diverse clades and the development of simple, quantitative theories linking temperature and oxygen availability to the extent of viable habitat in the oceans have enabled the use of Earth system models to link geochemical proxy constraints on environmental change with quantitative predictions of the amount and biogeography of habitat loss. Early indications are that the interaction between physiological traits and environmental change can explain substantial proportions of observed extinction selectivity for at least some mass extinction events. A remaining challenge is quantifying the effects of primary extinction resulting from the limits of physiological tolerance versus secondary extinction resulting from the loss of taxa on which a given species depended ecologically. The calibration of physiology-based models to past extinction events will enhance their value in prediction and mitigation efforts related to the current biodiversity crisis.

## Impact statement

Mass extinction events represent the greatest catastrophes in the history of animal life and only five major extinction events have occurred across the past 550 million years. Geological evidence can reveal the physical and chemical processes that caused environmental change, but differences in morphological, ecological, and physiological traits between extinction victims and survivors provide our best record of actual kill mechanisms. In recent years, this field has advanced both through the compilation of experimental data on organismal traits, enabling new insights into extinction patterns, and through the development of mechanistic models for biological response to environmental change, enabling incorporation of physiological tolerance into climate models to predict extinction patterns. Ultimately, mass extinction events are a critical source of data to calibrate the magnitude and rate of biological response to climate change over timescales longer than those of experiments and field studies. In this way, integration of information from the fossil record is becoming essential to the task of predicting and mitigating taxonomic losses due to current environmental change.

## Introduction

Earth is currently undergoing a biodiversity crisis on a scale unprecedented in the history of the human species (Barnosky et al., 2011; Dirzo et al., 2014; McCauley et al., 2015), but crises of similar or greater magnitude have occurred at least five times across the 600-million-year history of animal life (Figure 1A) (Raup and Sepkoski, 1982; Barnosky et al., 2011). All major mass extinction events are associated with evidence of rapid environmental change. In some cases, such as the end-Permian (252 million years ago [Mya]) and end-Triassic (201 Mya) mass extinctions, there is evidence for rapid and pronounced climate warming (Kiessling and Simpson, 2011; Payne and Clapham, 2012; Blackburn et al., 2013; Burgess et al., 2014; Bond and Sun, 2021). By contrast, the Late Ordovician (443 Mya) and Late Devonian (372 Mya) extinctions occurred in association with climate cooling (Joachimski and Buggisch, 2002; Finnegan et al., 2011). The

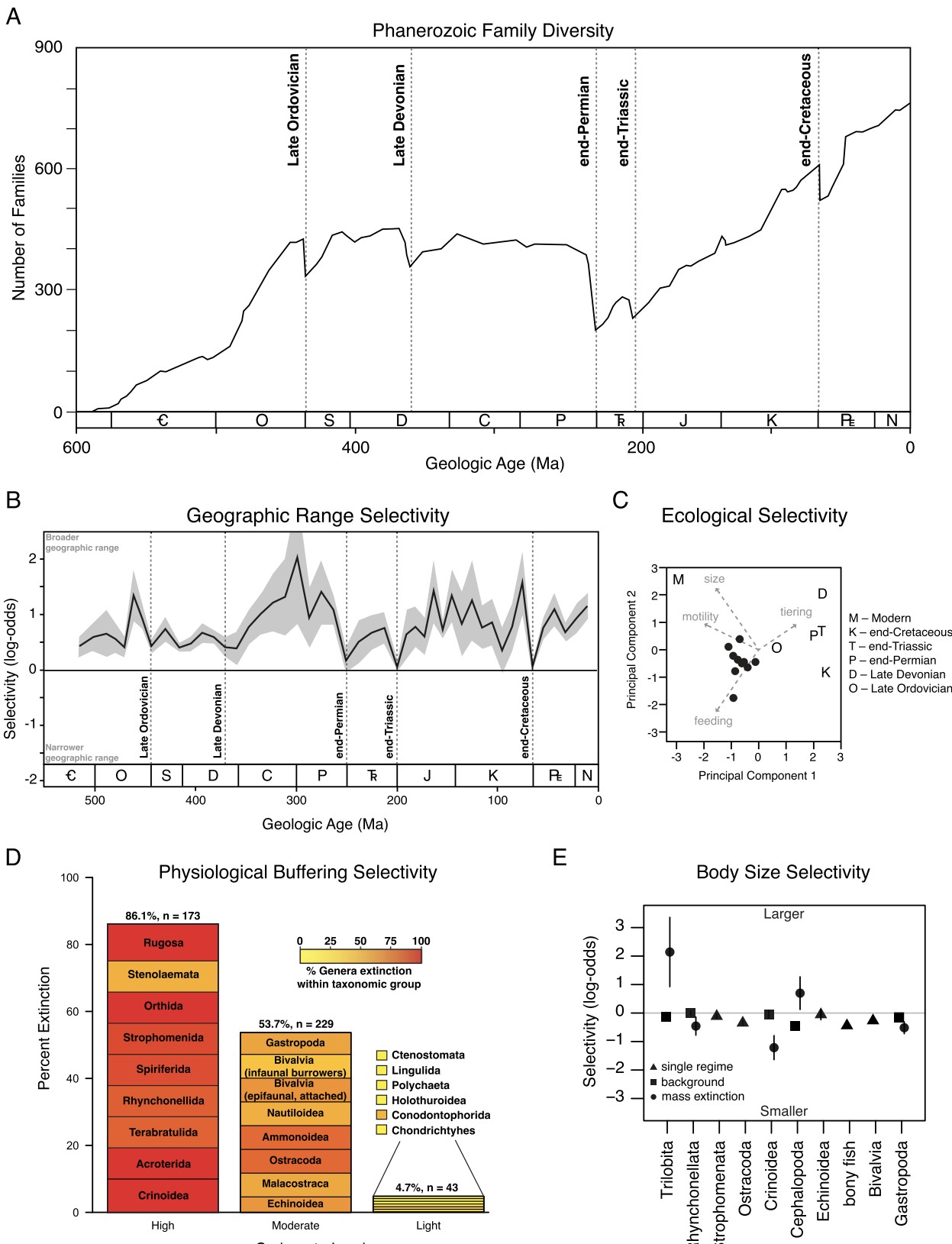

**Figure 1.** Extinction patterns in the fossil record. (A) Graph of marine animal diversity across the past 600 million years, illustrating the diversity declines associated with the five major mass extinction events (modified from Raup and Sepkoski, 1982). (B) Extinction selectivity with respect to geographic range, illustrating the preferential survival of broadly distributed genera during background intervals and the greatly reduced selectivity during mass extinction events (modified from Payne and Finnegan, 2007). (C) Principal components analysis of logistic regression coefficients of ecological traits and body size selectivity of the Big Five mass extinction events and the modern oceans, demonstrating the unique selectivity of the modern extinction threat (modified from Payne et al., 2016b). (D) Extinction selectivity during the end-Permian mass extinction, illustrating the preferential extinction of heavily calcified marine animal classes with less complex respiratory and circulatory systems (modified from Knoll et al., 2007; Knoll and Fischer, 2011). (E) Extinction selectivity with respect to body size for major classes of marine animals, illustrating the general bias of background extinction against smaller-bodied genera versus the variable direction of selectivity for classes that exhibit distinct patterns during mass extinction (modified from Monarrez et al., 2021).

end-Cretaceous extinction (66 Mya) was associated with an asteroid impact event whose aftermath resembled the consequences of a hypothetical global thermonuclear war (Pollack et al., 1983; Turco et al., 1983). Due to the magnitude and global scale of the current "Sixth" extinction, these events from Earth's past provide historical reference points for predicting the long-term magnitude, ecological impact, and recovery timescale from the current crisis or other, potential, human-mediated catastrophes.

While mass extinctions have been identified in the fossil record based largely on the magnitude of diversity loss across many higher taxa (Newell, 1963, 1967; Raup and Sepkoski, 1982), causal inference has relied more on geological and geochemical evidence of potential triggers (Alvarez et al., 1980; Svensen et al., 2009; Finnegan et al., 2011) and patterns of extinction selectivity interpreted to reflect proximal kill mechanisms (Jablonski, 1986; Sheehan and Hansen, 1986; Valentine and Jablonski, 1986; Jablonski and Raup, 1995; Knoll et al., 1996, 2007; Smith and Jeffery, 1998; Finnegan et al., 2012; Penn et al., 2018). Selectivity patterns have been assessed with respect to a wide range of traits (Figure 1B–E), including geographic range (Jablonski, 1986; Kiessling and Aberhan, 2007; Payne and Finnegan, 2007; Dunhill and Wills, 2015), body size (Jablonski and Raup, 1995; Friedman, 2009; Longrich et al., 2012; Allen et al., 2019; Payne and Heim, 2020; Monarrez et al., 2021), abundance (Lockwood, 2003; Payne et al., 2011), larval ecology (Valentine and Jablonski, 1986), diet (Wilson, 2013), functional ecology (Bambach et al., 2002; Payne et al., 2016b; Hughes et al., 2021), environmental breadth (Jablonski and Raup, 1995), respiratory and circulatory anatomy (Knoll et al., 1996, 2007; Clapham, 2017), and shell mineralogy (Clapham and Payne, 2011; Kiessling and Simpson, 2011).

Extinction selectivity provides our most direct evidence of proximal kill mechanisms (Raup, 1986), but to date, most testing of observed extinction patterns against hypothesized kill mechanisms has been semi-quantitative, focused on establishing consistency between predicted and observed directions of selectivity under various hypothesized kill mechanisms. Recently, advances in paleontological databases, geochemical proxies, physiological experiments, and Earth system and ecosystem models have enabled the comparison of observed and predicted extinction patterns within quantitative, self-consistent frameworks (Figure 2) (Penn et al., 2018). Although quantitative model-data comparison between observed and predicted extinction patterns is still in its early days, the door for direct comparison of past and future biotic response to climate change is now open, increasing the value of the fossil record in the mitigation of the current biotic crisis.

## Pattern

Analyses of selectivity for individual mass extinction events date back many decades (Jablonski, 2005). Studies synthesizing and comparing selectivity patterns across all major mass extinctions (and intervening background intervals) have emerged more recently, alongside publicly available databases of fossil occurrences and other traits (Alroy, 1999; Payne and Finnegan, 2007; Peters, 2008; Kiessling and Simpson, 2011; Payne et al., 2016b; Smith et al., 2018; Payne and Heim, 2020; Monarrez et al., 2021).

Geographic range is one of the traits most commonly hypothesized to correlate with extinction risk due to its influence on the extent to which populations of a given taxon may avoid a regional disturbance or have broad enough physiological tolerance limits or ecological capacities to survive a global one. Analyses of fossil data have confirmed that widely distributed taxa survive preferentially during background intervals (Figure 1C) (Jablonski, 1986, 2005; Payne and Finnegan, 2007). Broader geographic range is also significantly associated with survival during at least some major mass extinction events (Jablonski and Raup, 1995; Finnegan et al., 2016), but the strength of this association (i.e., the change in odds or probability of extinction per unit change in geographic range) is greatly reduced relative to background intervals (Figure 1C) (Kiessling and Aberhan, 2007; Payne and Finnegan, 2007). Due to the consistency of the association and the expectation of selectivity on total geographic range under most extinction scenarios, these patterns have rarely yielded direct insight into kill mechanisms. By contrast, the biogeography of extinction can be more informative. For example, end-Cretaceous echinoid extinction was significantly more severe in areas proximal to the Chicxulub impact site (Smith and Jeffery, 1998), and differences in extinction intensity across latitude often correspond with expectations due to climate change (Finnegan et al., 2012; Penn et al., 2018; Reddin et al., 2019, 2021). Quantifying the expected magnitude of spatial gradients in extinction intensity and differences in such gradients across higher taxa (or functional groupings) is the key to linking these findings with hypothesized kill mechanisms, and one that is already being partially realized (Penn et al., 2018).

The extinctions of large mammals during the Pleistocene (0.0117 Ma) and of large, non-avian dinosaurs during the Maastrichtian (66 Ma) have long prompted speculation that large-bodied animals are at systematically higher risk of extinction during times of environmental change (Raup, 1986; Wallace, 1889; Brown, 1995). Analyses of the fossil record reveal a more heterogeneous relationship, and one that may differ across taxa and habitats. For example, smaller body size is generally associated with greater extinction risk during background times for many classes of marine animals (Figure 1D) (Payne and Heim, 2020; Monarrez et al., 2021). By contrast, body size was not generally associated with extinction probability for terrestrial mammals until the Pleistocene (Alroy, 1999; Smith et al., 2018). End-Cretaceous extinctions preferentially eliminated larger-bodied fish, lizards, and snakes (Friedman, 2009; Longrich et al., 2012) but were unbiased in bivalves and gastropods (Jablonski and Raup, 1995). End-Permian extinctions preferentially affected larger foraminifera and brachiopods (Schaal et al., 2016). Many taxon-size combinations have yet to be examined systematically. In marine animals, size selectivity changes between background and mass extinction in many classes but the direction and magnitude of the size bias during mass extinction differs among classes (Figure 1D) (Payne and Heim, 2020; Monarrez et al., 2021). The differences in responses among classes remain to be explained. Because body size correlates with many ecological and physiological traits (Peters, 1983), size bias on its own is insufficient to diagnose proximal kill mechanisms but may be useful in conjunction with other traits or in testing against predictions of specific kill mechanisms (Deutsch et al., 2022).

Some mass extinction events exhibit selectivity patterns that can be mapped onto respiratory and circulatory anatomy, potentially reflecting underlying differences in susceptibility to metabolic stress from hypercapnia, anoxia, climate warming, or their interactive effects. For example, the end-Permian mass extinction preferentially affected heavily calcified marine animal genera with limited respiratory and circulatory systems (Figure 1B), suggesting a role for hypercapnia and/or direct and indirect fitness effects of acidification on shell dissolution (Calosi et al., 2017) in driving the extinction (Knoll et al., 1996). At the same time, the lack of sophisticated oxygen-supply mechanisms would also make these taxa more sensitive to temperature-dependent hypoxia (Deutsch

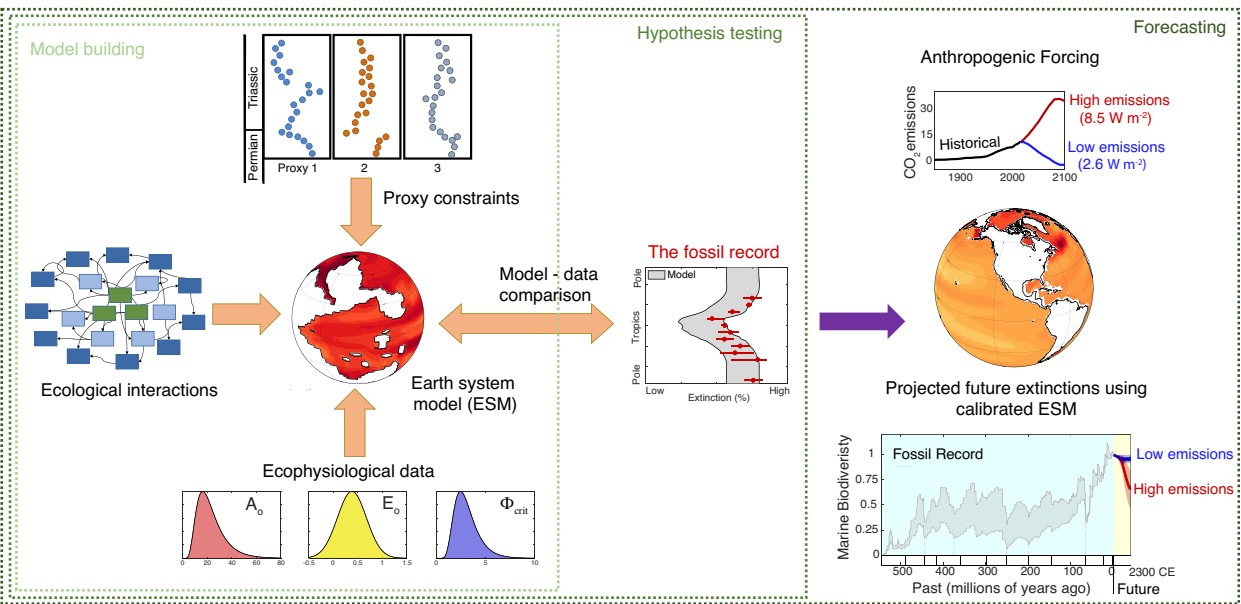

**Figure 2.** Workflow illustrating the use of geological and geochemical data to constrain Earth system models (ESMs), physiological experiments to constrain parameters used to populate models with species of different ecophysiotypes, and fossil occurrence data to conduct model-data comparison. Ecosystem structure remains to be incorporated into such models and can be used to predict extinction cascades. Calibration of models against selectivity patterns in ancient extinction events will improve their use in forecasting biotic response to current and future environmental change. Panels on right showing CO$_2$ emissions curves and future biodiversity projections are from Penn and Deutsch (2022).

et al., 2020; Endress et al., 2022) and metabolic differences among groups likely influence taxonomic selectivity patterns from changes in CO$_2$, temperature, and O$_2$. Similar patterns as seen in the end-Permian apply to other extinction events, including the end-Triassic mass extinction (Clapham, 2017; Kiessling and Simpson, 2011), consistent with shared kill mechanisms. By contrast, the end-Cretaceous mass extinction exhibits the opposite pattern, with taxa thought to be more sensitivity to ocean acidification surviving preferentially (Kiessling and Simpson, 2011), potentially reflecting differences in extinction patterns triggered primarily by volcanism versus impact events. The extent to which these patterns stand out from background extinction remains incompletely studied. A study controlling for differences between benthic versus planktonic and nektonic taxa indicates that many background intervals show the same selectivity, often of similar magnitude (Payne et al., 2016a). As discussed below, results of physiological experiments on living relatives of species in the fossil record are enabling quantitative prediction of biological response to past environmental changes inferred from geological and geochemical proxies. This is currently an area of rapid progress.

Simultaneous analysis of extinction selectivity across multiple traits and time intervals enables quantitative comparison of selectivity patterns between background and mass extinction as well as among mass extinction events (Figure 1E). Such analyses generally confirm that mass extinction events differ in selectivity from background patterns (Figure 1C, E) (Payne and Finnegan, 2007; Kiessling and Simpson, 2011; Finnegan et al., 2012; Payne et al., 2016b; Monarrez et al., 2021) and that the pronounced size bias of the modern extinction makes it an outlier relative to major mass extinctions as well as to recent background intervals (Figure 1D) (Payne et al., 2016b; Smith et al., 2018).

Overall, selectivity patterns accord with geological and geochemical data, indicating that mass extinction events are typically associated with large and rapid environmental perturbations rather than intensification of background extinction processes (Alvarez

et al., 1980; Hallam and Wignall, 1997; Finnegan et al., 2011). Testing hypothesized kill mechanisms requires simultaneous consideration of selectivity across multiple variables because physiological and ecological traits are often linked in complex ways. For example, body size is related to the supply and demand of oxygen (Deutsch et al., 2015, 2022) and food (Gearty et al., 2018) as well as to trophic level (Romanuk et al., 2011).

## Process

### Introduction

Understanding the causes of extinction selectivity in the fossil record requires additional information about the patterns of environmental change, the sensitivity of species to those changes, and disruptions in ecological networks. The interpretation of extinction selectivity thus relies on geochemical reconstructions of climate, understanding of the ecological and physiological traits of living taxa and, increasingly, on models that incorporate all these aspects of ecological and Earth system dynamics into an internally consistent, quantitative framework (Figure 2).

Patterns of extinction selectivity can arise simply from the fact that environmental changes can be highly variable in strength or even direction across space. Extinction selectivity could also arise from taxonomic or geographic differences in physiological sensitivity to environmental change, even if climate trends were globally uniform. In general, these factors are likely to be connected, as the tolerance limits of taxa to environmental conditions will shape the pre-extinction geographic distribution, which may confer greater or lesser sensitivity to environmental change in certain regions. Contemporary studies have advanced a mechanistic approach to investigating the causes of selectivity in mass extinctions by integrating many of these elements, from geochemical proxies of climate change, the modern diversity of ecophysiological traits, and the climate dynamics of Earth system models. In ocean studies,

emphasis has been on integrating climate and physiological constraints (Penn et al., 2018; Stockey et al., 2021). Terrestrial studies, by contrast, have tended to focus on ecological (food web) mechanisms largely missing from marine analyses (Roopnarine, 2006; Roopnarine and Angielczyk, 2015). These dichotomous approaches have made significant advances in their respective domains, paving the way for more unified marine and terrestrial studies.

### Example: Metabolic Index

One promising avenue for examining physiological kill mechanisms for ancient extinction events is the Metabolic Index, which was initially developed to test whether the biogeographic distributions of species are physiologically limited by $O_2$ supply and demand in the modern ocean (Deutsch et al., 2015). This ecophysiological model quantifies habitat viability for a species, in terms of its ability to carry out aerobic respiration, by taking a ratio of environmental oxygen supply to biological oxygen demand as a function of temperature and taxon-specific metabolic and $O_2$ supply traits (Eq. (1)). The metabolic energy demands of water-breathing marine animals increase with water temperature and body size (Gillooly et al., 2001), raising corresponding biological $O_2$ requirements. Temperature and body size also impact the rates of organismal $O_2$ supply through diffusion, ventilation, and internal circulation (Deutsch et al., 2022; Endress et al., 2022), while warmer water holds less ambient $O_2$. The ratio of temperature and body size ($B$)-dependent rates of potential $O_2$ supply and organismal metabolic demand, termed the Metabolic Index ($\phi$), quantifies the metabolic viability of a habitat for a given species:

$$\Phi = A_o B^\varepsilon pO_2 \, exp\left\{\frac{E_o}{k_B}\left[\frac{1}{T} - \frac{1}{T_{ref}}\right]\right\}, \qquad (1)$$

where $A_0$ (atm$^{-1}$) is the ratio of $O_2$ supply to resting demand rate coefficients, or hypoxia tolerance at a reference temperature and body size ($B$), with allometric scaling exponent $\varepsilon$ and Arrhenius temperature sensitivity, $E_0$ (eV), and $pO_2$ and $T$ are the oxygen partial pressure and temperature of ambient water, respectively (Figure 3) (Deutsch et al., 2015, 2020). These physiological traits and their distributions across taxa can be estimated from critical oxygen thresholds in respirometry experiments conducted for diverse marine biota over the past half century (Rogers et al., 2016; Chu and Gale, 2017). Critical oxygen thresholds define the Metabolic Index to be 1 (i.e., $\phi = 1$), allowing the traits to be estimated for organisms in a resting state under laboratory conditions. In the environment, $O_2$ requirements are elevated by more strenuous activities important for population persistence, such as growth, reproduction, feeding, defense, or motion. These additional energy demands require the $O_2$ supply to be raised by a factor, $\phi_{crit}$, corresponding to sustained metabolic scope (Peterson et al., 1990). Stable aerobic habitat barriers thus arise in ocean regions where the Metabolic Index falls below $\phi_{crit}$, while the geographic positions of these barriers depend on the species' traits (Deutsch et al., 2020). The habitability of any given parcel of water can therefore be determined from the temperature and oxygen partial pressure given the species values of $A_0$, $E_0$, and $\phi_{crit}$. Earth system models can be populated with hypothetical species by drawing combinations of values from the trait distributions (Figure 3). The promise of this framework for paleontological application is that trait distributions can be used to predict the patterns of biodiversity, providing a means for testing the model against the fossil record. Indeed, the observed tropical dip in marine species richness

observed for diverse animal groups in the modern ocean (Chaudhary et al., 2021) can be explained by aerobic habitat limitation implied by modern species Metabolic Index traits (Penn and Deutsch, 2022). Environmental temperature and oxygen concentration can be quantified using geochemical proxies for ancient events to calibrate Earth system models and body size can be measured from fossil specimens. In principle, ecological interactions can be further incorporated to model, allowing extinction cascades to be accounted for alongside direct, climate-driven habitat loss (Figure 4).

During periods of climate warming, rising water temperatures can drive the metabolic $O_2$ demand above a supply declining from ocean deoxygenation, leading to the loss of available aerobic habitat, and eventually species extinctions at local and global scales (Penn et al., 2018; Reddin et al., 2020). At regional scales, such as in the California Current System, aerobic habitat changes have been linked to multi-decadal fluctuations of anchovy populations, including near-extirpation of larvae from portions of their range (Howard et al., 2020). At global scales, aerobic habitat loss under the climate change simulated for the end-Permian mass extinction predicted a geographic selectivity of extinction consistent with the fossil record (Figure 5A): Extinction risk was greater for species inhabiting higher latitudes. This geographic selectivity arises because species previously occupying the tropics would already have been adapted to warm, low-$O_2$ conditions that became more widespread, whereas polar habitat niches disappeared more completely (Penn et al., 2018). In contrast to the geographic selectivity predicted for warming, periods of global cooling, such as during the Late Ordovician, are expected to generate extinctions focused on the low latitudes (Saupe et al., 2020), consistent with the patterns observed for that mass extinction (Finnegan et al., 2012) and may also occur through aerobic habitat loss if accompanied by deoxygenation (Finnegan et al., 2016) or due to declining hypoxia tolerance in cold water in species with thermal optima (Boag et al., 2018; Endress et al., 2022). Aerobic habitat loss is also predicted to select against large-bodied species, with a strong variability within size classes that depends on a species' temperature sensitivity (Deutsch et al., 2022). Extinctions driven by aerobic habitat loss may also explain the amplified background extinction rates observed for the early Phanerozoic, because of dramatically lower atmospheric $O_2$ levels and thus species living closer to their ecophysiological limits (Stockey et al., 2021). Trait adaption to different past climate states (Bennett et al., 2021) has the potential to buffer or amplify predicted extinction risks. The role of differences in ecophysiological traits across taxonomic groups in explaining observed patterns of extinction selectivity across higher taxa (Knoll et al., 1996, 2007) remains an open area of research.

Primary extinctions driven by the loss of aerobic habitat have the potential to be amplified by secondary extinctions arising from food web effects (Figure 4) or co-occurring environmental stressors that exacerbate direct aerobic habitat loss (Figure 5J–O). Aerobically tolerant species could still be lost if they are ecologically tied to vulnerable ones, for example, through the food web (Figure 4) or other critical interactions. Ocean acidification (Figure 5M–O) has the potential to further deplete aerobic habitat through direct $CO_2$ effects on critical oxygen thresholds, but the magnitude and direction of this effect is uncertain and variable across limited available experimental studies (Figure 3E) (Rosa et al., 2013; Lefevre et al., 2015). On its own, the magnitude of primary extinction from climate warming and associated physiological stresses depends on the amount of habitat loss beyond which a species can no longer sustain a viable population (i.e., the extinction threshold) (Urban,

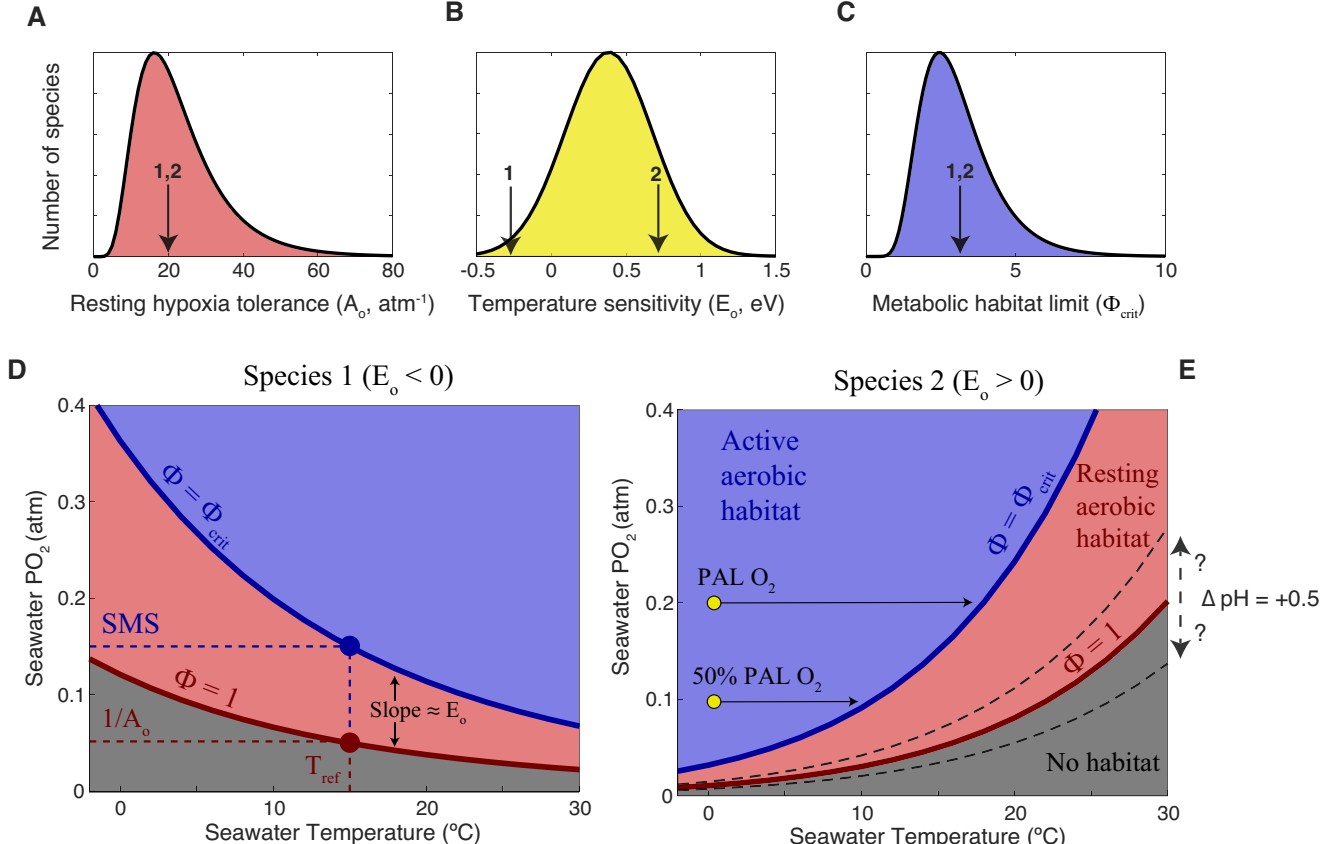

**Figure 3.** Graphs illustrating the key species traits of the Metabolic Index ($\phi$) along with how $\phi$ relates to temperature and oxygen partial pressure. (A–C) Frequency distributions of the Metabolic Index parameters for marine animals. (D, E) Graphs of variation in $\phi$ as a function of temperature and oxygen for species with negative (D) and positive (E) temperature sensitivities ($E_o$) of hypoxia tolerance ($A_o$), which is the inverse of the critical oxygen threshold (red circle) at a reference temperature ($T_{ref}$), as derived from respirometry experiments. For species in a resting state, the aerobic habitat limit occurs when $\phi = 1$, but in the environment a species' activity level or sustained metabolic scope (SMS) elevates the habitat limit to $\phi_{crit}$. For species with negative $E_o$, aerobic habitat availability increases with temperature, whereas for those with positive $E_o$ (i.e., most species; panel B), aerobic habitat declines with warming. Changes in $PO_2$ has the potential to lower aerobic habitat availability, and thus the amount of warming a species can withstand, as exemplified for two scenarios of with different fractions of present atmospheric levels of $O_2$ (PAL; yellow dots and arrows). A change in $CO_2$ also has the potential to alter hypoxia tolerance, but the magnitude and direction of this effect is unknown across marine biota and is illustrated here from experimental data for a single species under $\Delta$ pH = +0.5 (Rosa et al., 2013). Arrows in A–C denote species traits in D and E.

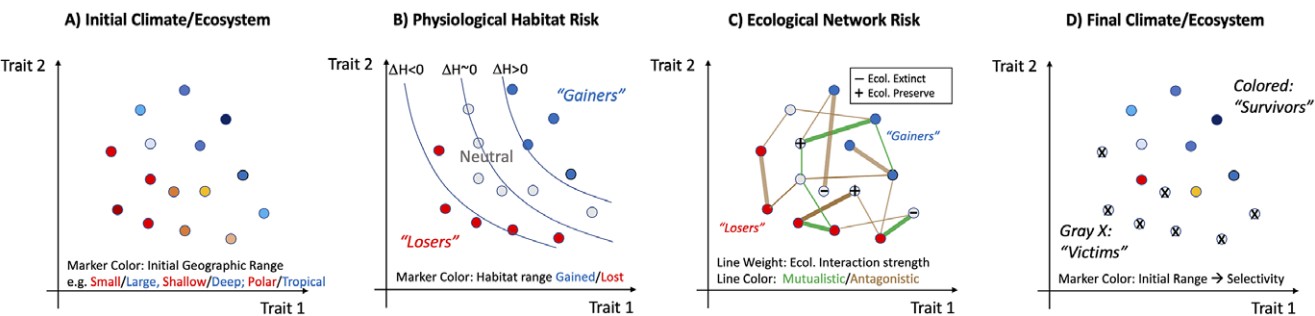

**Figure 4.** Hypothetical progression of a mass extinction highlighting sources of trait-based and geographic selectivity and potential ecological amplification. (A) An initial distribution of species (or "ecophysiotypes") defined by traits under selection by large-scale environmental conditions will likely result in systematic correlations between traits and geographic range. The range metric here can be considered overall range size (area and volume), or centroid (e.g., low-latitude versus high-latitude, shallow versus deep). (B) The initial biota are subjected to climate perturbation that poses a direct stress through a reduction in fitness whose magnitude depends on species traits and on local climate trends. The resulting change in available habitat ($\Delta$H; contours) presents an ecophysiological extinction risk that is geographically selective because it is trait selective (but may also be caused by climate patterns themselves). In this hypothetical case, habitat loss ($\Delta$H < 0) selects against species with high values of two traits (habitat "Losers") and may even benefit species with low values of those traits (habitat "Gainers"; $\Delta$H > 0). (C) Physiological extinction poses further ecological risks (or advantages) depending on the mutualistic or adversarial interactions with ecophysiotypes (nodes in graph) that are under trait-selective risk. Ecological risk is complex and for any particular species will depend on the physiological risk faced by the other species with which it interacts, which may be positive (green lines) or negative (brown lines), and strong (thick lines) or weak (thin lines). The results of these associations, which may be multiple and indirect, could alter extinction risk by either preserving ecological fitness ("+" symbol) or reducing it ("−" symbol). Changes in extinction risk are likely to be most pronounced for those in the neutral zone whose antagonists go extinct or who are buoyed by prey/mutualists that are under positive selection. (D) Post-extinction ecosystem, equal to the initial one (A) minus the ecotypes that have gone extinct from either primary (B) or secondary (C) effects.

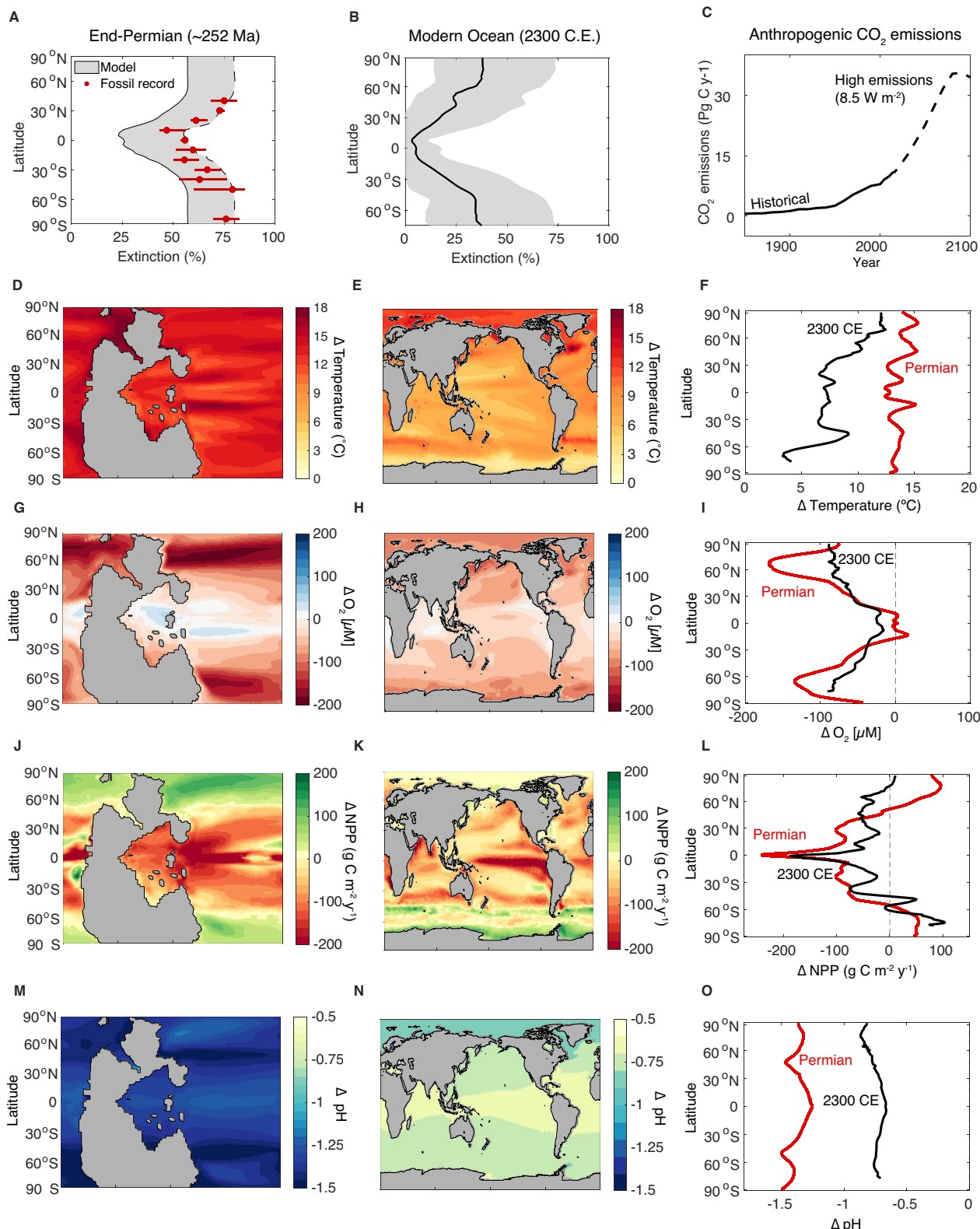

**Figure 5.** Geographic patterns of extinction and ocean changes in Earth system model simulations of the end-Permian climate transition (left column) and under anthropogenic greenhouse gas forcing (C) to 2300 C.E. (middle column). Line plot comparisons of end-Permian and potential future environmental changes versus latitude are shown in panels on the right (F, I, L, O). Model extinctions (A, B) are driven by ocean warming (D, E) and $O_2$ loss (G, H), as quantified through the Metabolic Index, and in (A) reproduce the latitudinal pattern from the fossil record of the end-Permian (red points). These primary extinctions have the potential to be amplified by other environmental stressors like changes in net primary productivity (NPP) (J, K) or pH (M, N) or through secondary extinctions via the food web. Shaded region in (A) shows uncertainty in end-Permian extinction magnitudes across a range of potential extinction threshold parameters. Solid line in (B) shows future extinction risk averaged across Earth system models, using an extinction threshold calibrated from the end-Permian (same as the solid line in A) (see Penn et al., 2022 for calibration details), while the shading in (B) shows the inter-model range. Future changes are projected under a high greenhouse gas emissions scenario, leading to a net radiative forcing of 8.5 W m$^{-2}$ in 2100 C.E. (C) and are relative to the pre-industrial era (1850–1900). Model fields are averaged over the upper 500 m, and for the future projections, they are averaged across Earth system models ($n = 5$). Model details are provided in Penn et al. (2018, 2022). Panels A–C are modified from Penn et al. (2018, 2022), respectively.

2015; Penn et al., 2018; Penn and Deutsch, 2022), even if population decline takes a long time to occur. Extinction thresholds may vary across species, but the average value at the global ecosystem level has been estimated from comparison of end-Permian model simulations to the fossil record, and assuming a similar loss of habitat that drove extinctions in the past would apply in the modern ocean (Penn and Deutsch, 2022). Calibration of this parameter from the fossil record has recently been used to project future extinction risk from climate changes resembling those of the end-Permian, which are arising today due to accelerating anthropogenic greenhouse gas emissions (Figure 5).

### Example: Food webs

Terrestrial paleo-community dynamics are usually modeled according to trophic ecology and body size to investigate the role of food-web topology in the propagation of disruptions caused by environmental change. Models of extinction cascades suggest that responses can be complex, resulting from both bottom-up and top-down effects (Kaneryd et al., 2012), with debate about whether simple or complex communities are more susceptible to such cascades and whether trophic versus other ecological interactions are most important (Eklöf and Ebenman, 2006; Donohue et al., 2017). Explicit consideration of extinction cascades during mass extinctions has generally focused on the consequences of collapse in primary production (Tappan, 1968; Vermeij, 2004). Bottom-up models predict extinction of smaller-bodied species in both the marine and terrestrial realms, due to the correlation of body size with trophic level, and exacerbated paleo-community instability post-extinction, which are consistent with investigations conducted on patterns of selectivity in relation to body size (Dunne et al., 2002; Roopnarine, 2006; Roopnarine et al., 2007; Dunne and Williams, 2009; de Visser et al., 2011; Lotze et al., 2011). Interestingly, the end-Cretaceous mass extinction, for which we have the strongest evidence for collapse of primary production, is associated with preferential extinction of larger-bodied species in some clades (Friedman, 2009; Longrich et al., 2012) but not with the preferential extinction of smaller-bodied species, suggesting that physiology or other ecological factors (including top-down extinction cascades) were important in determining survivorship.

Two challenges remain in the modeling of extinction via networks of ecological interactions. First, evidence that "primary" extinctions may often occur via environmental change that exceeds the physiological tolerance limits of species at many positions in the food web creates a need for further investigation of how food webs respond to such losses. Are extinction cascades more, or less, extensive when driven by primary extinctions occurring simultaneously at multiple trophic levels? Second, there is the challenge of integrating physiological and ecological models such that the full response of the marine or terrestrial ecosystem could be predicted in an integrated manner from the modeling of climate change to the loss of species that cannot physiologically tolerate the modified world, to the loss of species that depended on ecological interactions with species lost via primary extinctions (Figure 4). Differences in timescale and level of biological organization at which physiological and ecological processes dominate add to this challenge.

### Application to the sixth extinction

Mass extinction events provide our best source of information regarding the response of the biosphere to planetary-scale environmental disruption and the timescales and mechanisms of

subsequent recovery. This information may be particularly important for the oceans, where observing biological response to environmental change is challenging and where the fossil record is particularly complete and diverse. Since the industrial revolution, the oceans have experienced substantial changes in ocean biogeochemistry, mainly because of rapid injection of $CO_2$ into the atmosphere from anthropogenic sources. Under the accelerating future anthropogenic emissions scenario consistent with historical trends (Figure 5C), the oceans are expected to warm by 4–5°C and pH is expected to decrease, on average, by 0.44 pH units by the end of the 21st century, with changes increasing even further over the next few centuries (Figure 5E, N) (Kwiatkowski et al., 2020). High temperatures are also expected to reduce the ocean's oxygen content while also altering nutrient cycles (Sweetman et al., 2017). Unabated anthropogenic emissions could drive the oceans toward widespread oxygen deficiency over the rest of the 21st century and beyond (Figure 5H) (Breitburg et al., 2018).

Such changes would have drastic consequences for marine ecosystems as evident from declining fish stocks, expansion of marine dead zones, and reduced primary productivity across different parts of the globe (Figure 5K) (Blanchard et al., 2012). Efforts are already underway to project changes in species' ranges and abundances in response to climate change on land and in the oceans (Thuiller, 2004; Cheung et al., 2009; Chen et al., 2011; Pinsky et al., 2020). Extrapolating results from experiments and field observations over days or years to timescales of centuries, millennia, and beyond is challenging because different processes may dominate the biospheric response on different timescales, although there is emerging evidence that responses to some stresses are concordant across timescales (Reddin et al., 2020). Furthermore, the primary phase of extinction, dominated by physiology, may give way over time to a secondary phase of extinction, dominated by the effects of changing ecological interactions. Connecting the physiological and ecological processes driving extinction remains a research frontier.

Studies from the fossil record show that the ecophysiological constraints on marine taxa due to global warming and ocean deoxygenation will exert a key role in determining their risk to extinction under current and future emissions scenarios. The fossil record can even be used to calibrate the Earth system models used to predict future extinctions and changes in geographic range, just as paleoclimate records are used to calibrate models providing climate projections (Zhu et al., 2022). Under a high emissions scenario (Figure 5C), the marine biological richness could be reduced to 65% of its current state due to global warming and oxygen loss from oceans by 2,300 (Penn and Deutsch, 2022). The combined climate-ecophysiological models indicate that the local loss of species is expected to be the highest in tropical to temperate regions where taxa are expected to undergo a significant loss of aerobic habitat at their warm/low-$O_2$ range boundaries. In contrast, in terms of global habitat loss and extinction risk, the equatorial taxa are expected to fare better overall in low oxygen and warmer oceans compared to polar species due to their higher tolerance limits to warm climates and opportunities to expand their available habitats as the poles become more like the present-day tropics. This scenario has precedent in the fossil record with the end-Permian mass extinction where a similar latitudinal extinction pattern unfolded (Figure 5A, B) (Penn et al., 2018; Reddin et al., 2019). Further work to integrate the effects of changes in pH, $pCO_2$, salinity, and other key environmental variables into physiological performance models has the potential to make these models more general and accurate in reconstructing the causes of past extinction and predicting the consequences of future global change.

The ecological functions disrupted by global warming and marine defaunation are also bound to have cascading effects which could lead to extinction of vulnerable taxa. Modeling such effects is challenging due to the complexity of the interactions involved. The fossil record is our only source of data on the effects of major environmental disturbance at global scale. Fortunately, calibration of environmental change to physiologically expected extinction is becoming possible due to parallel advances in geochemistry, Earth system modeling, and physiological experimentation. The next decade will require integration of food webs and other types of ecosystem models to extract the full value of the lessons from Earth's past in forecasting and guiding its future.

**Open peer review.** To view the open peer review materials for this article, please visit http://doi.org/10.1017/ext.2023.10.

**Data availability statement.** No data were collected or analyzed as part of this review paper.

**Acknowledgments.** The authors thank G. Wilson Mantilla, P. Calosi, C. Rasmussen, and an anonymous reviewer for their constructive feedback on the manuscript.

**Author contribution.** All authors contributed to the conceptualization, original draft preparation, review, and editing of the manuscript.

**Financial support.** This work was supported by the National Science Foundation (J.L.P., grant number EAR-2121392; C.D., grant number EAR-2121466).

**Competing interest.** The authors declare no competing interests exist.

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
