## [Reviewer Report]

*Comments to Author*: Review of the article by Payne et al. entitled “Selectivity of mass extinctions: patterns, processes, and

future directions”

The work by Jonathan L. Payne, Jood A. Al Aswad, Curtis Deutsch, Pedro M. Monarrez, Justin L. Penn and Pulkit Singh is really interesting y, and will make a significant contribution to the research field of extinctions, but also to the fields of global change ecology and biology. However, the article requires some work before to be ready for publication. Particularly, provide a more balanced discussion of the existing evidence and limitations of the approach proposed, verify semantics in few cases, and consider to integrate existing references and concept I highlight below. I believe you can relatively easy deal with the points I raise, and whilst the comments may lead to minor or major revision. I selected major only to enable you have more time if you require so.

Major Comments

Line 149-151. In my opinion, here, you are only skimming the surface. You correctly mention the simple Bauplans of heavy calcified marine invertebrates can in part explain the extinction patterns described in the fossil record for the end-Permian. The point is valid and interesting. I could argue, however, that Bauplan “simplicity” can be an advantage, enabling an organism to be less impacted directly by (at least some) environmental drivers, and indirectly possess the advantage to have lower metabolic-energetic requirements. In this sense, organisms with more complex, high-performance cardio-circulatory and respiration, systems may possess greater homeostatic abilities (see for example Melzner et al. 2009 for ocean acidification), but they will not be able to sustain high homeostatic activities for extended periods of time. Complexity requires more energy for maintenance and repair. Modern ecosystem collapse, such as in the North Sea following a switch of a keystone copepod species (from more to less nutritional), the loss in high level predators was documented. In addition, you seem to (I say this respectfully and with the best of intention to help you further improving your excellent work) miss the importance of the effects of other drivers, which can act along different pathways. For example, here specifically, you are not discussing of the direct (i.e. corrosion of carbonated structure) and indirect (e.g. increase in energetic cost for compensatory calcification to maintain a positive net mineralisation in light of passive dissolution) effects of ocean acidification conditions on marine heavy calcifiers. Please ref. to the literature on this topic. It is vast, but it needs to (even if synthetically) be acknowledged, as it is super relevant. I can refer to my own work to help, but the choice of ref. to use sits entirely with you: please see Calosi et al. 2019 Ann. Rev. Mar. Sc. for the discussion of putative mechanisms-pathways of action (see summary Figure 4), and to Calosi et al. 2017 Nat. Comms. and Findlay et al. 2011 Mar. Biol. Res. where differences in passive corrosion-net mineralisation in different population of the same species and different species are reported. but as I suggest, the literature on this topic is extensive.

Line 193. I feel that in the paper, “ecological-ecosystemic” aspects of mass extinction should be discuss even more thoroughly. The authors do so, but more thinking and work should go into this to push the work and the MS to where it can, and deserved to be. Ensure to bring this as forward as possible in the MS. The method you propose is great to paint a picture on the direct (physiological) impacts of temperature and oxygen (not all drivers characterising current global change for example) but cannot be used to depict the indirect (ecological) impacts. You recognise the issue, but this can be stated even more clearly. You need to discuss in greater depth how best the tool you propose can be used, to what advantage, and where it cannot be used or used recognising the presence of limitations. In line 280 and following you could integrate the work by Reddin et al. 2020 NCC (which you cite) to the discussion, related to early extinctions being caused primarily by direct physiological effects, and later extinctions being caused by indirect ecological effects. All considered, I suggest to enhanced the discussion, and I fully recognise the authors’ effort in the intellectual integration of “physiological” and “ecological” extinctions …. and figure 4 is excellent by the way!!

Line 110, 203-204, 280 and elsewhere. Please the first time you mention “tolerance” and “resilience” ensure to provide a definition in parenthesis: 110 and 204, respectively. I have the impression that in most cases you use it “correctly” but in other not (ex. line 282. Tolerance is the ability of a biological system to resist change following or under a perturbation, and resilience is the ability of a biological system to come back to the original stable state it was in after a perturbation (i.e. resilience is the inverse of the time needed to get back to the starting status).

Line 266. Pre-adaptation does not exist. It is an erroneous concept. An organism cannot be adapted to something it has not experienced before. And if it is adapted to be able to face a certain challenge, it is because it has already experienced it in his phylogenetic history, or it has been selected to be tolerant to another stressor which grants it “protection” to the stressor we are investigating. In either of these cases, we talk of “exaptation” (i.e. existing adaptation), please see Gould’s seminal work on this topic.

For your interest, we have recently published a work on the diversity and evolution of thermal limits, testing for the presence of paleo signals on these traits, and we found one. It was not the strongest of signals of the evolutionary drivers we investigated, but it was non the less significant. In short, the era of emergence of a taxonomic group has a significant influence of defining its thermal limits. See Bennet et al. 2021 Nat. Comms. “The evolution of critical thermal limits of life on Earth”.

Line 287-289. Consider to integrate here and elsewhere some discussion of the importance of extreme and intense, but temporary, climatic-environmental events. These are primary drivers for local and regional extinction, helping shaping physiological diversity (see for example (1) for the CEH Bozinovic et al. 2011 Ann. Rev. Ecol. Evol. Syst. and references within, and (2) for extreme drivers as evolutionary driver for thermal limits Bennet et al. 2021 Nat. Comms.) participating to global extinctions. I think for example to Marine Heat Waves, intense salinity changes, intense upwelling events, etc.

Minor Comments

Line 19. Specify “differences” in what to make the statement explicit and clearer to readers.

Line 20. << .. has advanced .. >>

Line 26. Add after << … in this way, .. >> something along the line of << ..the integration of information emerging from .. >>

Line 26. Change << … the ask ... >> to something more descriptive << .. to help us in the arduous task .. >>

Line 31, 46, 158, 306 and elsewhere. Give Latinisms in Italics << .. versus .. >>.

Line 128-145. Here consideration to the fact that not the same mechanisms often apply to terrestrial and aquatic (marine-freshwater) organisms should be given. Oxygen for example is 20 times lower in water than in air. The same point must be considered for body plans and body plans’ complexity, which differ greatly between terrestrial and aquatic organisms. This applying not only across phyla, but also within phyla, see for example body plan and physiological differences between terrestrial (reptiles, birds, mammals) and aquatic chordates (fish, and secondarily marine mammals), and also between terrestrial (insects and arachnids) and aquatic (crustaceans).

Line 153. “bake”?

Line 157. Unclear what << .. poorly buffered taxa … >> means. Do you mean << … organisms with poor homeostatic abilities .. >>?

Line 190-191. This is obvious! Conceptually tautological.

Line 242. What about intra and interspecific competition?

Line 320-329. I see the challenge here to integrate data that have a different “grain”, considering that physiological and ecological processes occur at different time scales, and are studied at different time scales and level of biological complexity.

Line 341. Incorrect terminology. We cannot say that seawater become more acidic, in fact it is less alkaline, but even so it is always best to refer to the phenomenon of “ocean acidification” or to talk about “a reduction in pH”, but to avoid to say “more acidic seawater”. It would be semantically incorrect. Also note that a reduction in pH it is but one of the symptoms of ocean acidification, as the increase in seawater pCO2 and the reduction in carbonate ions and omega values are also important element to mention. Finally remember that 0.44 an estimate for the global ocean, locally and regionally conditions vary greatly.

Line 351-355. I am not convinced of this point. Based on physiological abilities, rare species should be sensitive (more sensitive) in the early stage of extinction based on the fact that they possess narrower physiological windows of tolerance: based on extant organisms’ physiology of course, as we cannot define empirically that of extinct species. To provide an example, I have shown, and I am not the only one, that there is a thermal limit physiology of rare versus common species (Calosi et al. 2008 J. Biogeog., Calosi et al. 2008 Biol. Letts.), and that the breadth and central position of the latitudinal range of extent of modern species, as well as their southern and northern most geographical limits, are predicted by the breadth of their thermal window and their CTmax and CTmin, respectively. I bring this as an example among many.

However, I do not know whether in secondary phases of extinction, what were before rare versus common species, are more or less favoured.

All consider the term “long-term” confuses me, as I think here it is more relevant to state in which “phases” of the extinction we are: the early or primary stage when direct physiological effects are more relevant, or the later or secondary phases when indirect ecological effects matter more? Consider to change the discussion in this direction. Apologise if the “binary” view of an extinction I used here does not make justice to a far more-complex phenomenon.

Line 363-379. I agree fully with the statement, but what about other major drivers? Ocean acidification, changes in salinity, etc?

More in general your discussion only focuses on temperature and oxygen, and whilst I recognise the primary impacts of these major drivers, they are not the only one, and they do not occur in isolation: see for example works by Côté, Piggot, Carrier-Belleau and others on multi-stressors occurrence and non-linear effects on aquatic organisms.

---

## [Reviewer Report]

*Comments to Author*: Review of the manuscript ”Selectivity of mass extinctions: patterns, processes, and future directions” submitted to Cambridge Prisms: Extinction by Payne et al.

This manuscript seeks to address mainly intrinsic processes governing the selectivity of mass extinction events. The manuscript is nicely written and in clear language. There are many good examples highlighting the different mass extinctions and I think therefore that this manuscript will serve as an excellent instalment within the field of mass extinctions as it reviews past research highlighting many relevant references in the field in a nice way while at the same way calls out specific directions within this research field that may of particular relevance for the current Anthropocene biodiversity crisis. 

While I do not have any ‘objections’ to anything written I do have a few comments that the authors might find relevant. Firstly, it stumbled upon the statement in the abstract (l. 34 and again l. 108–126) that ‘geographic range matters less during mass extinction’. While this may be true, I just note that in the case of the Late Ordovician crisis, we do actually see a lot of selectivity with respect to geographic ranges (see, for instance, Finnegan et al., 2016, Proc. Roy. B, http://dx.doi.org/10.1098/rspb.2016.0007). This reference would also be relevant for lines 268–272 (aerobic habitat loss).

l. 132. Regarding body size, it could perhaps be of relevance to throw in a few references on the ‘lilliput-effect’ during mass extinctions. From the top of my head, I recall Huang Bing had a paper on the end-Ordovician and Richard Twitchett one on the end-Permian, but there are quite a few more. 

Also, perhaps of relevance to the discussion in l. 222–388 (latitudinal extinction patterns), I think the authors might find a new preprint by Ontiveros et al: ‘Cooling Oceans Did Trigger Ordovician Biodiversifcation’ (it should be on Research Square) interesting as it models the latitudinal biodiversity gradient during the Middle Ordovician cooling, showing how cooling climate affected biodiversity accumulation positively (i.e. the opposite scenario than what is seen during extinctions where warming reduces biodiversity levels).

Minor pedantic edits:

l. 20: add ‘d’ after ‘advance’ ->’advanced’

l. 153: change ‘b’ to ‘m’ so that it reads ‘make’ instead of ‘bake’ :-)

To sum up, I enjoyed reading this very comprehensive and interesting manuscript and I, therefore, encourage publication almost as is.

-Christian Rasmussen.

---

## [Reviewer Report]

*Comments to Author*: This is a concise and well-written review discussing patterns of extinction selectivity in the fossil record. A particular emphasis is placed on mass extinctions and how fossil data--when combined with Earth system models--may help to provide constraints on likely patterns of extinction stemming from anthropogenic global change. The paper provides an admirable overview of a diverse field and condenses the topic into a manageable size perfect for a general audience, student seminars, and so on. My only suggestions to the authors concern a few minor edits (issues of consistency, typos, parts that the authors themselves have flagged as needing an update) indicated below. I look forward to seeing the final version of this contribution.

Line 20: advance -> advanced

Line 31: changing -> change

Line 56: be sure to include social media summary

Line 153: bake -> make

Line 192: “marine” is indicated here, but is the approach potentially more general?

Line 212-213: condense this line. could read “a way for more unified marine and terrestrial studies” or similar.

Line 232: replace “-” with “:” given that it sets off a list of items?

Line 241 (and elsewhere): check that “2” is in subscript for O2

Line 399: replace “), (” with “;”

Lines 686, 689: subscript “2” for O2

Line 701: write out “low latitude vs high latitude”

Figure 1, panel B: Chondrichtyes -> Chondrichthyes

Figure 1 (and elsewhere): use shared conventions for axis labels; currently a mix of first word capitalized, all words capitalized, no words capitalized

---

## [Editor Report]

*Comments to Author*: Dear Dr. Payne and coauthors, 

Thank you for your submission on selectivity of mass extinctions. It was an enjoyable read and will make a fine contribution with some minor revision. Of the three reviewers, two had very minor comments, mostly typological that should be easy to address. Reviewer #1 had more substantial comments reflecting minor disagreement on some points, clarification on others, and additional references that should be considered for citation to better reflect the literature. I encourage you to carefully consider their comments. With this last point about literature citation, I would agree that in some sections (the K/Pg terrestrial section, an area that I study) the coverage of the literature could be increased. I don’t mean to suggest that you need to cite my work but there are a number of relevant papers on extinction selectivity across the K/Pg in terrestrial biota (Field et al. 2018 Curr Biol; Hughes et al. 2021 Ecol & Evol; Wilson 2013 Paleobiology) that you should consider to bolster your survey of the literature. 

We look forward to your revisions so that we can move this contribution to accepted. 

Sincerely,

Greg Wilson Mantilla

---

## [Reviewer Report]

*Comments to Author*: Thank you for addressing previous comments. I caught only a few issues here, all of them very minor and easily sorted:

Throughout: there are variable abbreviations with the same meaning (Ma, Mya).

Line 141: fish -> fishes.

Line 165: sensitivity -> sensitive.

Lines 352-3: is it redundant to indicate oceans experienced change to “ocean biogeochemistry”?

---

## [Editor Report]

*Comments to Author*: Dear Authors, 

Thank you for the careful revisions made to your manuscript “Selectivity of mass extinctions: patterns, processes, and future directions.” We are happy at this time to accept this excellent contribution to the journal. The reviewer made a few very minor suggested corrections that we would like you to address before publication. Again, we greatly appreciate the attention to your revision process and the excellent resulting product. 

Sincerely,

Greg Wilson Mantilla